# Taxonomy of the Genus *Bryobia* Koch (Acari: Tetranychidae): Reconsideration of Subgenera and Updated Species Groups

**DOI:** 10.3390/insects15110859

**Published:** 2024-11-03

**Authors:** Jawwad Hassan Mirza, Fahad Jaber Alatawi, Muhammad Kamran, Carlos Holger Wenzel Flechtmann

**Affiliations:** 1Department of Plant Protection, College of Food and Agriculture Sciences, King Saud University, Riyadh 11451, Saudi Arabia; jmirza@ksu.edu.sa (J.H.M.); murafique@ksu.edu.sa (M.K.); 2Departamento de Entomologia e Acarologia, Escola Superior de Agricultura “Luiz de Queiroz”, Universidade de São Paulo, Piracicaba 13418-900, Brazil; chwflech@usp.br

**Keywords:** *Allobia*, associated setae, *Bryobia*, dendrogram, dorsocentral setae, duplex setae, species groups

## Abstract

The subgeneric divisions of the genus *Bryobia* are hereby revised, based on literature, to recognize three subgenera, i.e., subgenus *Bryobia* Koch (duplex setae present on leg tarsi III–IV), subgenus *Allobia* Livschits and Mitrofanov (duplex setae absent on leg tarsi III–IV) and subgenus *Lyobia* Livschits and Mitrofanov (duplex setae present only on leg tarsus III). The species in each subgenus is further categorized into three species groups based on the position of the fourth pair of dorso-central setae *f*1. Taxonomic discussion is provided for eight *Bryobia* species with ambiguous tribe status and two species are considered *species inquirendae*. Furthermore, a key to the subgenera and species group is also provided. The present study is an effort to solve the taxonomic inconsistencies in the genus *Bryobia* and simplify the identification of its species.

## 1. Introduction

The genus *Bryobia* Koch [1] is one of the oldest genera in the family Tetranychidae and the largest in the subfamily Bryobiinae [2,3]. It comprises 149 described species reported from all over the world [3]. Some members of this genus are considered significant pests and have quarantine importance, including *B*. *praetiosa* Koch (the type of the genus), *B*. *rubrioculus* (Scheuten), *B*. *graminum* (Schrank), and *B*. *kissophila* (Eyndhoven) [4]. The clover mite, *B*. *praetiosa*, is a notorious member of the genus, infesting different economic fruit, grain, and ornamental crops and is distributed worldwide [3,4].

The genus *Bryobia* belongs to the tribe Bryobiini and can be readily identified by the uncinate true claws and a pad-like empodium [5]. The diagnosis of the genus *Bryobia* was recently updated and morphologically differentiated from the closely related genera *Toronobia* Meyer and *Pseudobryobia* McGregor based on the number of tenent hairs on leg empodia II–IV and the coxal setal formula [6]. However, the taxonomy of the genus itself is complicated and has been claimed to be challenging [2,5,7]. It was primarily due to inconsistent and variable morphological characters that differentiate the *Bryobia* species. This resulted in synonymies of different species over time [2,8]. In addition, the larval stages were considered to differentiate among species, which did not help much [5,7,9]. The genus was divided into seven species groups to solve the issue related to species complexes [10]. Later, two subgenera were erected as *B*. (*Bryobia*) and *B*. (*Pseudobryobia*) [11]. Livschits and Mitrofanov [12] raised the status of *Pseudobryobia* as a genus and separated it from the genus *Bryobia* based on the number of setae on coxae II. They also proposed five subgenera in the genus *Bryobia*, viz, *Bryobia*, *Lyobia*, *Allobia*, *Periplonobia*, and *Eharobia* [12], while the sixth subgenus was added by Mitrofanov [13] as *Bryobiopsis*.

To date, many of the *Bryobia* species remain unassigned to any of the subgenera via their original designation. Furthermore, the existing subgenera of the genus *Bryobia* were not followed by different researchers where either the diagnostic characteristics of the subgenera were misinterpreted [14,15] or researchers disagreed about using any rank or classification under the genus level [16]. Furthermore, some species described after 1973 share diagnostic morphological characteristics of at least two existing *Bryobia* subgenera (*sensu* Livschits and Mitrofanov). This further raises concern about the subgeneric diagnostic characteristics and the overall subgeneric divisions. Adding to the problem are those *Bryobia* species that do not follow the diagnosis of the tribe Bryobiini [7,16]. The recent work of Arabuli et al. [6] reviewed the genus *Pseudobryobia*, recognized the six subgeneric divisions of Livschits and Mitrofanov [12] and Mitrofanov [13] in the genus *Bryobia*, and transferred seven species of the *Pseudobryobia* to the subgenera *Allobia* (three species) and *Bryobiopsis* (four species). In light of this, the described *Bryobia* species should also be reviewed taxonomically.

The molecular-based phylogeny has been scarcely studied for the *Bryobia* species. Only around 300 sequences are available on the GenBank database, which includes molecular sequences amplified by different markers for seven morphologically identified *Bryobia* species, focused only on genetic analysis for some *Bryobia* species delineation or inclusion as outgroups [17,18,19,20,21,22]. Even after such studies [17,20,22], the phylogenetic relationship of *Bryobia* species remained unresolved.

Nevertheless, a misidentified species could be assigned to an incorrect taxon, which could lead to a dilution of the overall diagnostic characteristics of that taxon. The present study aimed to acknowledge this scenario in the genus *Bryobia* by providing a detailed analysis of the morphological characteristics of diagnostic significance at the subgeneric and species level, re-evaluating the existing subgeneric divisions and species groups, proposing updated taxonomic rankings and species groups, and providing notes on the status of some doubtful species.

## 2. Materials and Methods

### 2.1. Taxonomic Characterization

The taxonomic literature of 149 *Bryobia* species was critically reviewed to confirm the current status of subgeneric divisions. The diagnostic characters of each existing subgenus were thoroughly understood in all described species. The species groups were also created and a key was prepared to identify each rank. In addition, the status of eight *Bryobia* species published by Meyer [7,16] with intermediate characteristics of the tribes Bryobiini and Hystrichonychini were discussed. The strength of each morphological characteristic was evaluated for its suitability at the subgeneric level based on the persistence of subgeneric diagnostic characteristics in each described species. The nomenclature of Lindquist [23] was followed for body setae and leg chaetotaxy. The term “duplex present” used in the present study, refers to the association of the sensory solenidion and a tactile seta on tarsi III and IV, while “duplex absent” refers to dissociated setae, where sensory solenidion and its tactile seta are distantly present.

### 2.2. Phylogenetic Analysis

The assignment of *Bryobia* species in the proposed taxonomic ranks (i.e., three subgenera) was analyzed by constructing phylogenetic trees based on both morphological characteristics and COI sequences submitted to the NCBI GenBank database.

The 14 morphological characters used in the dendrogram included those distinct characters which have been previously used in erecting subgeneric divisions by Livschits and Mitrofanov [12] and Mitrofanov [13] (Appendix A) and those provided in the updated generic diagnosis by Arabuli et al. [6] (Appendix A). Furthermore, three species of the genus *Pseudobryobia*, *P*. *bakeri* McGregor [24], *P*. *canescens* Baker and Tuttle [25], and *P*. *curiosa* [26], were also considered in the morphology-based dendrogram. The morphological characteristics for the subgeneric evaluation were analyzed via TNT (tree analysis using new technology) vers. 1.6 [27] with four algorithms, i.e., sectorial search [28], drift [28], Ratchet [29], and tree fusion [28], with standard settings. The designated out–group was *Pseudobryobia bakeri*, and there were 1000 replicates for the bootstrap and the jackknife.

The molecular analysis was accomplished by acquiring genetic sequences of the COI gene of *Bryobia* species only from the GenBank database “https://www.ncbi.nlm.nih.gov/nuccore (accessed on 28 October 2024)”. It included accession numbers EU487091 (*B*. *praetiosa* strain NL10), EU487092 (*B*. *kissophila* strain FR13), EU487095 (*B*. *berlesei* strain SP7), EU487112 (*B*. *rubrioculus* strain GR12), EU487125 (*B*. *sarothamni* strain FR16) [17], MF278615 (*B*. *abyssiniae*; [21]), and OQ064101 (*B*. *polymorpha* isolate Massane-2) [15]. The morphological identity of these sequences was confirmed by those authors who submitted the sequences to the database and has been mentioned in the publications [15,17,21]. There is no genetic sequence from any gene/marker submitted for *Pseudobryobia* species on the GenBank database. Hence, the species *Oligonychus afrasiaticus* (McGregor) (accession number DQ656482) was used as an outgroup in the molecular analysis. We conducted the maximum likelihood (ML) (SH-aLRT support %, aBayes parametric test, and ultrafast bootstrap support %) analysis and obtained branch support with the ultrafast bootstrap [30] implemented in the IQTREE 2 software with 1000 ultrafast bootstraps. using the best-fit model selected by IQTREE, TIM + F + G4, chosen according to BIC.

## 3. Results and Discussion

### 3.1. Taxonomic Background the Genus Bryobia

The systematics of the genus *Bryobia* is complex and confusing. It is because, in addition to morphological features of adults, immature stages, host plant associations, and geographic distributions were the factors considered while describing species or synonymizing them [8,9,12]. Koch erected this genus in 1836 based on the type species *B*. *praetiosa* and described three more species by 1838. In 1842, Koch provided the diagnosis of the genus. Unfortunately, no type specimens were preserved and the descriptions of those previously three published species were reported to be different from their illustrations [2,9,24].

From the 1840s to the 1930s, different species were described and later synonymized due to inconsistencies in the morphological characteristics with Koch’s *B*. *praetiosa*. McGregor [24] made an attempt to revise the genus *Bryobia* and erected the genus *Pseudobryobia*. Pritchard and Baker [2] synonymized the genus *Pseudobryobia* with *Bryobia*. Later, the genus was divided into seven species groups: *borealis*, *speciosa*, *praetiosa*, *rubrioculus*, *berlesei*, *sarothamni*, and *drummondi* [10]. These species groups were proposed based on the number and state of setae on leg femora I, the shape of the peritremes, and the shape of the spermathecae. Wainstein [11] disagreed with the synonymy of *Pseudobryobia* and considered it a subgenus of *Bryobia*.

Livschits and Mitrofanov [12] studied the taxonomy and systematics of the genus *Bryobia* in detail. The authors considered a lot of parameters for the taxonomic revision of the genus, including the morphology, number of generations (univoltine, bivoltine, etc), host plant association, host plant type (annual, biennial, etc), geographic distribution, etc. Out of 89 species described up to that time, 39 species were assigned to five subgenera (*Bryobia* (12 species), *Lyobia* (12 species), *Periplonobia* (four species), *Allobia* (10 species), and *Eharobia* (one species), with 13 species as considered as doubtful and the remaining (about 37 species) synonymized. The morphological characteristics for subgeneric differentiation, including the developed state of the prodorsal lobes, the position of setae *c3*, the position of setae *f*1, the distance of *f*1–*f*1 vs. *f*2–*f*2, the presence or absence of duplex setae on tarsus IV, and the dorsal anterolateral angulations [12]. Following similar criteria, Mitrofanov [13] described the sixth subgenus, *B*. (*Bryobiopsis*) (one species), while two more species, one each for subgenera *Allobia* and *Periplonobia*, were described. With the six subgenera until 1973, 42 species were described under the genus *Bryobia*. Smiley and Baker [31] considered the genus *Septobia* Zaher et al. [32] (with only two species) as a subgenus of the *Bryobia*, which was later synonymized with the genus *Bryobia* [5]. Appendix A summarizes the morphological features of the six subgenera after Livschits and Mitrofanov [12] and Mitrofanov [13].

Today, among the 149 described taxa of the genus *Bryobia*, about 33 species were originally designated to a certain subgeneric rank. The rest of the species remain without a subgeneric assignment as proposed by Livschits and Mitrofanov [12]. Additionally, even after the subgeneric ranks of Livschits and Mitrofanov [12] and Mitrofanov [13], *Bryobia* species were described under certain species groups (including the *berlesei* species group and the *praetiosa* species group) as previously proposed by Eyndhoven [10]. The inconsistencies with these species groups are the same as with the subgeneric divisions, i.e., most authors did not follow them while describing new species. Meyer [16] suggested avoiding trinomial nomenclature for the *Bryobia* species and identifying the African species without assigning them to any sub-rank. It is important to mention that Meyer [16] followed Pritchard and Baker [2] for the characteristics of tribes in the subfamily Bryobinae. According to this, the tribe Bryobiini could have pad–like true claws and empodium. The widely accepted classification of Bolland et al. [5] differentiates the tribe Bryobiini from the tribe Hystrichonychini based on the characteristics true claw uncinate and pad-like, respectively. This highlights the need to verify the taxonomic position of the species described by Meyer [7,16].

### 3.2. Morphological Inconsistencies in the Existing Subgenera of the Genus Bryobia

In our investigation of existing subgeneric divisions of Livschits and Mitrofanov [12] and Mitrofanov [13], there are considerable variations in the morphological characteristics, i.e., the state of the prodorsal dorsal lobes, the state of lateral angulations, etc. Sometimes, the characteristic states of two or more subgenera can be found in one species. Furthermore, some characteristic states become difficult and confusing to identify, leading towards incorrect subgeneric assignment and false identification of *Bryobia* species. There are >30 species that present morphological characteristics of two or more subgenera (Appendix A). For example, the deep incision characteristic state of the prodorsal lobes has been chiefly not well described or properly illustrated, leaving the characteristic ambiguous. Similarly, the position of setae *f*1 and comparative distance of *f*1–*f*1 vs. *f*2–*f*2 induced confusion related to the subgeneric assignment. Appendix A explains these ambiguities related to the diagnostic characteristics of the existing subgenera. Therefore, we agree with Meyer [16] that a classification system based on the concept of Livschits and Mitrofanov [12] and Mitrofanov [13] could complicate the taxonomy of *Bryobia*, the largest Bryobine genus. The recent revision of the genus *Pseudobryobia* further enhanced our understanding of the significance of diagnostic morphological characteristics [6]. They, however, agreed with the subgeneric divisions of Livschits and Mitrofanov [12] and Mitrofanov [13], particularly for the subgenera *Allobia* and *Bryobiopsis* [6], and provided diagnostic features of these subgenera.

Considering the taxonomic ambiguities related to the subgeneric ranking of the *Bryobia* species, the subgenera and their diagnostic morphological characteristics should be re–evaluated. Before the updated subgeneric divisions can be discussed, the diagnostic characteristics based on which Livschits and Mitrofanov [12] and Mitrofanov [13] added six subgenera are hereby discussed.

#### 3.2.1. The Development State of Prodorsal Idiosomal Lobes

All three subgenera, i.e., *Bryobia*, *Lyobia*, and *Eharobia*, were described as having well-developed dorsal lobes. The number of dorsal lobes could be three to four, where the species with three lobes were considered an aberration from the normal four lobes [6]. The subgenera *Bryobia* and *Lyobia* could be separated from *Eharobia* due to deep incisions between the outer and inner lobes (lobes developed in *Eharobia* separated without deep incisions). The interpretation of the development of lobes and the incisional separation of outer and inner lobes is unclear. Furthermore, the state of the prodorsal lobes could vary not only within the sexes (male and female) of a species but also among its developmental stages. Furthermore, this characteristic state has been considered an aberration in a few species [5,6,21]. Similarly, Fashing et al. [21] reported the characterisitic of the prodorsal lobes as a variation, called the differences in the number of lobes an aberration, and considered it as an intraspecific variation based on molecular analysis in *B*. *abyssiniae* Fashing et al. [21].

Furthermore, the species *B*. *coatesi* Meyer [7] was not originally assigned to any subgenus, and its lobe incisions were not described (Figure 1). The outer lobes are separated by deep incisions from the inner median single lobe, as illustrated. This could be assumed by observing the deep incisions of the species *B*. (*L.*). *gushariensis* Livschits and Mitrofanov (originally assigned to the subgenus *Lyobia*). It is important to mention that the subgenus *Lyobia* is characterized by well-developed lobes with outer lobes separated by deep incisions from the inner lobes (Appendix A). If we follow this concept, *B*. *coatesi* could be placed in the subgenus *Lyobia*. However, the original authors stated that *B*. *coatesi* lobes resemble that of *B*. *geyeri* Meyer (characterized as shallow incision). In that case, it should be assigned to the subgenus *Allobia*. Hence, it is unresolved.

Similarly, another example is of *B*. *pandayi* Eyndhoven and Vacante, whose lobe shapes are described, but the state of incision is not described nor illustrated. In the species *B*. *dianthi* Mitrofanov and Sharonov, the original authors designated this species to the subgenus *Lyobia*. However, the state of its lobes clearly depicts that the incisions are not deep.

#### 3.2.2. Dorso-Lateral Propodosomal Dorsal Angular Projections

The characteristic of lateral angulation is poorly described in different species (Appendix A). This characteristic was described as lateral angulation or shoulder-like projection/bulges present dorsally at the level above the eyes only to diagnose the species of subgenera *Bryobia* and *Eharobia sensu* Livschits and Mitrofanov. It is unclear what systematic purpose this characteristic serve, but a few species have been described and illustrated with this feature. In some instances, it is illustrated but not described, which could indicate that this characteristic is of low significance. It is also interesting to mention the species *B*. (*B*.) *cagani* Çobanoglu et al. [14], which was assigned to the subgenus *Bryobia* in the original description, even without lateral angulations. The authors did not provide any explanation for this assignment.

#### 3.2.3. Distance and Position of Setae *f*1 and *f*2 and Position of Setae *c3*

The subgenus *Eharobia* was uniquely described with setae *c_3_* vertically/longitudinally aligned to setae *c_2_* (Appendix A). This characteristic was combined with lobes developed without deep incisions and the presence of lateral angulations. However, as in Appendix A, there are species that do not follow this combination of characteristics while holding *c3* vertically aligned with *c2*. A similar situation is with the position of setae *f*1 and the relative position of *f*1 vs. *f*2 (Appendix A). For example, the species *B*. *abyssinae* Fashing and Ueckermann was not assigned to any subgenus (*sensu* Livschits and Mitrofanov), but on the basis of the position of setae *f*1 and the distance of *f*1–*f*1 vs. *f*2–*f*2, it could be assigned to either *Allobia* subgenus or *Periplonobia* (*sensu* Livschits and Mitrofanov). There are other such examples explained in Appendix A.

### 3.3. Subgenera of the Genus Bryobia

The taxonomic work done by Livschits and Mitrofanov [12] and Mitrofanov [13] was undoubtedly comprehensive and detailed. As time passed, the increased number of described species warranted the need to revise their concepts. As the confusion in the morphological features between the genera *Bryobia* and *Pseudobryobia* was previously resolved [6,12] and based on the ambiguities among the *Bryobia* subgenera exposed in the present study, we can confidently propose an updated subgeneric ranking to classify the species of the genus *Bryobia*. Among all the morphological characteristics used to erect subgeneric divisions, the absence or presence of duplex setae on tarsus III–IV remained consistent among all the *Bryobia* species and strikingly clear in all descriptions and illustrations. Based on this, three subgenera are proposed based on the absence or presence of a duplex on leg tarsus III and IV.

#### 3.3.1. Subfamily Bryobiinae Berlese

**Diagnosis:** Empodium with tenent hairs, females with three pairs of anal setae, and males with five pairs of genito-anal setae.


**Tribe Bryobiini Reck**


**Diagnosis:** True claws uncinate and empodium pad-like.


**Genus *Bryobia* Koch**


**Type species**: *Bryobia praetiosa* Koch 1836:9.

**Diagnosis (Based on females):** The genus *Bryobia* can be separated from the closely related genus *Pseudobryobia* based on the following combination of characteristics: coxal formula 2–1–1–1, empodium I with one or more than one pair of tenent hairs, dorsal propodosomal lobes with varying development (i.e., lobes are well developed where outer lobes are separated from inner lobes by a deep incision or lobes are poorly developed or absent), and the fourth pair of dorsal setae *f*1 varying in position from marginal to central.

There are a total of 149 species described in this genus [3], which are now divided into three subgenera i.e., subgenus *Bryobia* Koch, subgenus *Lyobia* Livschits and Mitrofanov, and subgenus *Allobia* Livschits and Mitrofanov. The species with a duplex present on both tarsi III–IV are transferred to the subgenus *Bryobia*, while those with a duplex present only on tarsus III are placed in the subgenus *Lyobia*. The species with a duplex absent on both leg tarsi III–IV are assigned to the subgenus *Allobia*. The other subgenera, *Bryobiopsis* and Eharobia (diagnosed due to the absence of duplex/associated setae), are synonymized with subgenus *Allobia* in the present study. There is no fixed pattern (absence or presence) of duplex setae on leg tarsi III–IV in the species of the subgenus *Periplonobia sensu* Livschits and Mitrofanov, i.e., species may or may not have duplex setae of those leg segments. Hence, its species are assigned accordingly to recognized subgenera in this study.

#### 3.3.2. Characteristic State of Solenidion and Tactile Setae on Tarsus III and IV

It is important to clarify the terminologies of duplex setae, associated setae, and dissociated setae. These three terms have been alternatively used in the literature, i.e., by Livschits and Mitrofanov [12], Mitrofanov [13], and Arabuli et al. [6]. This characteristic is significantly important for the updated subgeneric division in the present study. The terms “duplex setae” and “associated setae” are considered synonyms of each other in the present work, stating that on leg tarsus III and/or IV, the solenidion is basally close to the tactile setae such that a duplex is formed (Figure 2). If both the setae (solenidion and the tactile setae) are placed at a distance from each other (Figure 3), then it is considered that the duplex is absent.

### 3.4. Phylogenetic Relationship of the Proposed Subgeneric Divisions

A morphology-based dendrogram and COI–based cladogram are presented in Figure 4 and Figure 5, respectively. The characteristics and their states used for morphological analysis are provided in Appendix A.

#### 3.4.1. Morphological Analysis

The phylogenetic tree, constructed after analyzing 14 generic and subgeneric diagnostic characteristics, depicts that the phylogenetic relationship of genera *Pseudobryobia* and *Bryobia* is unclear (Figure 4). These genera have amalgamated taxonomic history [6] and were differentiated mainly due to the absence or presence of prodorsal lobes [5]. The detailed taxonomic discussion provided by Arabuli et al. [6] emphasized the significance of the number of setae on leg coxae. The morphological characteristics (Appendix A) used in the current analysis also included the characteristic state of the number of setae on coxae II, based on which the species of both genera could be clearly distinguished. As presented in Figure 4, two species of the genus *Pseudobryobia* were grouped together with the *Bryobia* species with 100/100 values for the bootstrap and the jackknife. However, there was poor or no support for the other sub-clades/branches (Figure 4).

Similarly, the grouping of the proposed subgenera in the genus *Bryobia* was also not supported due to low bootstrap and jackknife values (Figure 4). The monophyly or paraphyly of the genus *Bryobia* could not be assessed. The morphological characteristics used in the present study were of generic and subgeneric significance and have been previously been used to develop taxonomic ranks. It appears that to clearly understand the morphology-based phylogenetic relationship, the scope of analysis should be extended to many taxa and include a large set of characteristics and characteristic states.

#### 3.4.2. Molecular Analysis

The maximum likelihood (ML) tree constructed from COI sequences received from NCBI–GenBank, representing seven *Bryobia* species, did not support the proposed subgeneric divisions of the *Bryobia* genus (Figure 5). This finding is consistent with the previous studies where the phylogenetic relationship of different *Bryobia* species was reported as largely unresolved [17,20,22]. In the present study, the clade containing *B*. *abyssiniae*, *B*. *praetiosa*, *B*. *kissophila*, *B*. *sarothamni*, *B*. *polymorpha*, and *B*. *berlesei* was supported with a SH-aLRT support percentage, a Bayes parametric test, and ultrafast bootstrap values 87.6, 0.943, and 62, respectively (Figure 5). The sub-clade, including species assigned to the proposed subgenus *Bryobia* (*B*. *praetiosa*, *B*. *abyssiniae*, and *B*. *kissophila*), was poorly supported (63/0.6/51, Figure 5). However, the tree branch of *B*. *praetiosa* and *B*. *abyssiniae* was strongly supported (84.9/0.954/68, Figure 5). Based on 28S rDNA and COI sequences, the phylogenetic relationship of *B*. *kissophila* and *B*. *praetiosa* was poorly supported [17,20,22], as presented in the present study (Figure 5).

The sub-clade containing *B*. *berlesei*, *B*. *polymorpha*, and *B*. *sarothamni*, assigned to the proposed subgenus *Lyobia*, did not receive any support from ML analysis (12.5/0.436/37, Figure 5). However, the branch of *B*. *polymorpha* and *B*. *sarothamni* was strongly supported (93.1/0.999/77, Figure 5). Previous ML studies also showed the poor phylogenetic relationship between *B*. *sarothamni* and *B*. *berlesei* [17,20,22]. The species *B*. *rubrioculus* was located at the base of the clade containing *Bryobia* species without any support. These findings are consistent with previous phylogenetic analyses of 28S rDNA and COI, where *B*. *rubrioculus* is located distantly from other *Bryobia* species [17,22].

### 3.5. Subgenus Bryobia Koch

**Type species**: *Bryobia praetiosa* Koch

**Diagnosis** (based on females): Duplex setae on tarsus III–IV present. The tactile seta and the associated solenidion closely present (Figure 2). A total of 53 species were assigned to this subgenus. These species are further categorized into three species groups based on the position of *f*1 setae.

#### 3.5.1. *praetiosa* Species Group

**Exemplar species:** *Bryobia praetiosa* Koch

**Diagnosis (based on females):** Fourth pair of dorsocentral setae *f*1 present laterally, along the margin, and the distance *f*1–*f*1 always greater than *f*2–*f*2 (Figure 6a).

This species group is comprised of 43 species.*B*. *agioriticus* Hatzinikolis and Emmanouel [40]*B*. *alberensis* Auger and Migeon in [15]*B*. *attica* Hatzinikolis and Emmanouel [41]*B*. *bakeri* (Zaher, Gomaa & El-Enany) [32]*B*. *batrae* Baker and Tuttle [42]*B*. *borealis* Oudemans [43]*B*. *cagani* Çobanoğlu, Ueckermann and Cilbircioğlu [14]*B*. *cyclamenae* Hatzinikolis and Panou [44]*B*. *emmanoueli* Hatzinikolis & Panou [44]*B*. *exserta* Wang [45]*B*. *fuegina* Gonzalez [46]*B*. *geigeriae* Meyer [7]*B*. *gigas* Auger, Arabuli and Migeon [47]*B*. *glacialis* Berlese [48]*B*. *graminum* (Schrank) [49]*B*. *hengduanensis* Wang and Cui [50]*B*. *japonica* Ehara and Yamada [51]*B*. *karooensis* Meyer [7]*B*. *kissophila* Eyndhoven [52]*B*. *latisetae* Wang [45]*B*. *macrotibialis* Mathys [53]*B*. *magallanica* Gonzalez [46]*B*. *meteoritica* Hatzinikolis and Panou [44]*B*. *montana* Mitrofanov [13]*B*. *monticola* Wang [45]*B*. *neopraetiosa* Meyer [7]*B*. *neoribis* Tuttle and Baker [54]*B*. *nikitensis* Livschits and Mitrofanov [55]*B*. *nitrariae* He and Tan [56]*B*. *platani* Hatzinikolis and Panou [57]*B*. *praetiosa* Koch in [1]*B*. *qilianensis* Ma and Yuan [58]*B*. *qinghaiensis* Ma and Yuan [58]*B*. *reckiana* Mitrofanov and Strunkova [59]*B*. *strunkovae* Mitrofanov [60]*B*. *tadjikistanica* Livschits and Mitrofanov [61]*B*. *urticae* Sayed [62]*B*. *vasiljevi* Reck [63]*B*. *xiningensis* Ma and Yuan [58]*B*. *xizangensis* Wang [45]*B*. *yunnanensis* Ma and Yuan [58]*B*. *ziziphorae* Strunkova and Mitrofanov [37]*B*. *watersi* Manson [64]

#### 3.5.2. *osterloffi* Species Group

**Exemplar species:** *Bryobia osterloffi* Reck

**Diagnosis (based on females)**: Fourth pair of dorsocentral setae *f*1 present sublaterally where the distance *f*1–*f*1 is shorter than *f*2–*f*2 (Figure 6b).

This species group is comprised of seven species.*B*. *abyssiniae* Fashing and Ueckermann [21]*B*. *artemisiae* Bagdasarian [65]*B*. *burkei* Meyer [16]*B*. *osterloffi* Reck [66]*B*. *petrilunara* Meyer [16]*B*. *serifiotica* Hatzinikolis, Papadoulis and Kapaxidi [67]*B*. *variabilis* Manson [64]

#### 3.5.3. *neoephedrae* Species Group

**Exemplar species:** *Bryobia neoephedrae* (Gutierrez and Bolland)

**Diagnosis (based on females):** fourth pair of dorsocentral setae *f*1 present centrally, aligned with other three pairs of dorsocentral setae (Figure 6c).

This species group is comprised of only one species.*B*. *neoephedrae* (Gutierrez and Bolland) [6]

#### 3.5.4. Species Not Assigned to Any Species Group

There are two species, viz, *B*. (*B*.). *calida* (Karg) [68] and *B*. (*B*.). *lagodechiana* (Reck) [63], which belong to the subgenus *B*. (*Bryobia*) as proposed in the present study. However, these species were not described or illustrated with the characteristic state of setae *f*1–*f*2 and are not assigned to any species group.

### 3.6. Subgenus Allobia Livschits and Mitrofanov

*Bryobiopsis sensu* Mitrofanov [13] **synonym nov.**

*Lyobia sensu* Livschits and Mitrofanov [12] **synonym nov.**

*Eharobia sensu* Livschits and Mitrofanov [12] **synonym nov.**

**Exemplar species:** *Bryobia pritchardi* Rimando

**Diagnosis (based on females):** Duplex (associated) setae on tarsus III–IV absent. The tactile seta is present at a distance from the closest solenidion (Figure 3), prodorsal lobes variably developed (from developed but not deeply incised to minute lobes or lobes absent). A total of 28 species are included in this subgenus. These species are further categorized into three species groups based on the position of *f*1 setae.

#### 3.6.1. *pritchardi* Species Group

**Exemplar species:** *Bryobia pritchardi* Rimando

**Diagnosis (based on females):** Fourth pair of dorsocentral setae *f*1 present laterally, along the margin, and the distance *f*1–*f*1 is always greater than *f*2–*f*2 (Figure 6d).

There are 24 species included in this species group.*B*. *aegyptiacus* (Zaher, Gomaa & El-Enany) [32]*B*. *angolensis* Meyer [16]*B*. *beaufortensis* Meyer [69]*B*. *caricae* Hatzinikolis and Emmanouel [70]*B*. *coatesi* Meyer [7]*B*. *geyeri* Meyer [7]*B*. *giannitsensis* Hatzinikolis and Panou [44]*B*. *imbricata* Meyer [7]*B*. *incana* Meyer [69]*B*. *kakamaensis* Meyer [16]*B*. *lucens* Meyer [7]*B*. *macedonica* Hatzinikolis and Panou [44]*B*. *marcandrei* Hatzinikolis and Panou [44]*B*. *meyerae* Zaher, Gomaa & El-Enany [32]*B*. *monechmae* Meyer [7]*B*. *nigromontana* Meyer [69]*B*. *orycustodia* Meyer [39]*B*. *pritchardi* Rimando [38]*B*. *relhaniae* Meyer [69]*B*. *rhodesiana* Meyer [7]*B*. *spinescens* Meyer [7]*B*. *triloba* Meyer [7]*B*. *tuberosa* Meyer [7]*B*. *ylikiensis* (Hatzinikolis and Emmanouel) [71]

#### 3.6.2. *abbatielloi* Species Group

**Exemplar species:** *Bryobia abbatielloi* (Smiley and Baker)

**Diagnosis (based on females):** Fourth pair of dorsocentral setae *f*1 present centrally, aligned with other dorsocentral setae, and the distance *f*1–*f*1 is always shorter than *f*2–*f*2 (Figure 6e).

There are two species included in this species group.*B*. *abbatielloi* (Smiley and Baker) [31]*B*. *querci* Hatzinikolis and Panou [57]

#### 3.6.3. *deserticola* Species Group

**Exemplar species:** *Bryobia deserticola* Meyer

**Diagnosis (based on females):** fourth pair of dorsocentral setae *f*1 present sublaterally, neither aligned with other dorsocentral setae nor present marginally, and the distance *f*1–*f*1 could be shorter or longer than *f*2–*f*2 (Figure 6f)

There are two species included in this species group.*B*. *birivularis* Meyer [39]*B*. *deserticola* Meyer [39]

### 3.7. Subgenus Lyobia Livschits and Mitrofanov

**Exemplar species:***Bryobia rubrioculus* (Scheuten)

**Diagnosis (based on females):** Duplex (associated) setae present on tarsus III and absent on tarsus IV. The tactile seta is far from the closest solenidion on tarsus IV. A total of 59 species are included in this subgenus. These species are further categorized into three species groups based on the position of *f*1 setae. The species *B*. *confusa* Livschits and Mitrofanov was not assigned to any species group due to the position of setae *f*1 and *f*2 being neither described nor illustrated.

#### 3.7.1. *eurotiae* Species Group

**Exemplar species:** *Bryobia eurotiae* Mitrofanov

**Diagnosis (based on females):** Fourth pair of dorsocentral setae *f*1 present centrally or subcentrally, usually aligned with other dorsocentral setae, and the distance *f*1–*f*1 always shorter than *f*2–*f*2 (Figure 6g).

There are three species included in this species group.*B*. *ericoides* Meyer [7]*B*. *eurotiae* Mitrofanov [13]*B*. *pamirica* Mitrofanov [13]

#### 3.7.2. *sarothamni* Species Group

**Exemplar species:** *Bryobia sarothamni* Geijskes

**Diagnosis (based on females):** Fourth pair of dorsocentral setae *f*1 present sublaterally and the distance *f*1–*f*1 either shorter or longer than *f*2–*f*2 (Figure 6h).

There are eight species included in this species group.*B*. *annatensis* Manson [64]*B*. *chrysocomae* Meyer [7]*B*. *cinereae* Auger and Migeon [72]*B*. *nasrvasensis* Bagdasarian [73]*B*. *perinsignis* Eyndhoven and Vacante [34]*B*. *polymorpha* Auger and Migeon [15]*B*. *sarothamni* Geijskes [74]*B*. *spica* Pritchard and Keifer [8]

#### 3.7.3. *rubrioculus* Species Group

**Exemplar species:** *Bryobia rubrioculus* (Scheuten)

**Diagnosis (based on females):** Fourth pair of dorsocentral setae *f*1 present laterally and the distance *f*1–*f*1 always longer than *f*2–*f*2 (Figure 6i).

There are 47 species included in this species group.*B*. *aetnensis* Vacante [75]*B*. *alveolata* Auger and Flechtmann [76]*B*. *angustisetis* Jakobashvili [77]*B*. *astragali* Strunkova and Mitrofanov [37]*B*. *baroni* Auger, Arabuli and Migeon [78]*B*. *berlesei* Eyndhoven [79]*B*. *belliloci* Auger, Arabuli and Migeon [47]*B*. *bucharica* Strunkova and Mitrofanov [37]*B*. *cavalloroi* Vacante and Eyndhoven [80]*B*. *centaureae* Livschits and Mitrofanov [33]*B*. *cerasi* Hatzinikolis and Emmanouel [70]*B*. *chongqingensis* Ma and Yuan [58]*B*. *convolvulus* Tuttle and Baker [35]*B*. *cooremani* Eyndhoven and Vacante [34]*B*. *dekocki* Eyndhoven and Vacante [34]*B*. *dianthi* Mitrofanov and Sharonov [36]*B*. *dikmenensis* Eyndhoven and Vacante [34]*B*. *dubinini* Bagdasarian [73]*B*. *eharai* Pritchard and Keifer [8]*B*. *gushariensis* Livschits and Mitrofanov [32]*B*. *hadizeni* Barbar, Parker and Auger [81]*B*. *kakuliana* Reck [82]*B*. *kassioticus* Hatzinikolis and Panou [57]*B*. *livschitzi* Mitrofanov and Strunkova [59]*B*. *longisetis* Reck [66]*B*. *lonicerae* Reck [82]*B*. *mercantourensis* Auger and Migeon [72]*B*. *mirmoayedii* Khanjani, Gotoh and Kitashima [83]*B*. *nothofagi* Gonzalez [46]*B*. *obihsaphedi* Mitrofanov [60]*B*. *oblonga* Livschits and Mitrofanov [61]*B*. *pandayi* Eyndhoven and Vacante [34]*B*. *parietariae* Reck [66]*B*. *pelerentsi* Eyndhoven and Vacante [34]*B*. *piliensis* Hatzinikolis and Emmanouel [40]*B*. *populi* Wang and Zang [84]*B*. *provincialis* Eyndhoven and Vacante [34]*B*. *pyrenaica* Eyndhoven and Vacante [34]*B*. *rubrioculus* (Scheuten) [85]*B*. *siliquae* Hatzinikolis and Emmanouel [70]*B*. *strombolii* Vacante [75]*B*. *syriensis* Barbar and Auger [86]*B*. *tiliae* (Oudemans) [87]*B*. *ulicis* Eyndhoven [88]*B*. *ulmophila* Reck [66]*B*. *vandaelei* Vacante [75]*B*. *vaneyndhoveni* Vacante [75]

The remaining four species are either poorly described or illustrated, or the aspect of duplex/associated setae on tarsus IV is not provided (Appendix A). Those four species were not assigned to any subgenus and are as follows:*B*. *apsheronica* Khalilova [89]*B*. *desertorum* Hassan, Afifi and Nawar [90]*B*. *ribis* Thomas [91]*B*. *weyerensis* Packard [92]

### 3.8. Notes on Bryobia Species Described by Smiley and Baker [31]

The results of the present study and the revision of *Pseudobryobia* by Arabuli et al. [6] emphasize the significance of the number of coxal setae on leg II for the differentiation of the genera *Bryobia* (coxa II with one seta) and *Pseudobryobia* (coxa II with two setae). Arabuli et al. [6] transferred the species *B*. *abbatielloi* from *Pseudobryobia* to *Bryobia* stating that the diagnosis of both the genera by Smiley and Baker [31] were inversely stated, of what was established for both genera. Considering this concept, the two species (*B*. *tuttlei* and *B*. *pseudorubrioculus*) described as new in the genus *Bryobia sensu* Smiley and Baker [31] should also be classified in the genus *Pseudobryobia* from *Bryobia*, based on coxal setation. Both these species have two setae on coxa II following the genus *Bryobia* diagnosis by Smiley and Baker [31]. However, these two Bryobine species in question do not fit with the *Pseudobryobia* diagnosis too, due to the prodorsal lobes and setae *f*1 in lateral/marginal position. Previously, an attempt was made to accommodate an intermediate *Pseudobryobia* species, *P*. *nikitensis*, by erecting the genus *Nuciforaella* (Vacante), which was later synonymized with *Bryobia* (*Allobia*) [6]. In the given scenario, the genus *Nuciforaella* could be argued for reinstatement, but only with an updated diagnosis to accommodate the two Bryobine species. It is crucial to investigate the types of *B*. *tuttlei* and *B*. *pseudorubrioculus* to resolve the ambiguity in the generic diagnostic characteristics in these species. Unless that happens, these species are considered as *species inquirendae*.

### 3.9. Notes on the Eight Bryobia Species Described by Meyer [7,16]

Pritchard and Baker [2] separated the subfamilies of the family Tetranychidae, i.e., Bryobiinae Berlese and Tetranychinae, based on the absence or presence of empodium with tenent hairs and the number of anal setae in the female and male. A similar concept was followed by Bolland et al. [5]. However, the diagnostic differentiation of tribes in the subfamily Bryobiinae was a little contrasting. According to Pritchard and Baker [2], the tribe Bryobiini was characterized by having four pairs of dorsal prodorsal setae and 12 pairs of hysterosomal setae with true claws as curved hooks or long pads. This diagnosis of the tribe Bryobiini, followed by Meyer [7,16], was “True claws uncinated or if pad-like, prodorsum with well-developed setiferous lobes, empodium pad-like”. Later, Bolland et al. [5] updated this characteristic to include only those spider mites having true claws uncinated in the tribe Bryobiini and those mites having true claws pad-like in the tribe Hystrichonychini.

Interestingly, Meyer [7,16] described eight *Bryobia* species (Appendix A) with true claws pad-like/straight on only leg I, while legs II–IV had true claws uncinate. At the same time, Meyer [7] described two species with empodia and true claws pad-like on all legs. To accommodate those two species, the author erected the genus *Bryocopsis*, calling it an exception in the tribe Bryobiini. This genus resembled the *Bryobia* in its major morphological aspects but differed only in the state of true claws, as mentioned. According to Bolland et al. [5] and Guiterrez [93], *Bryocopsis* belonged to the tribe Hystrichonychini due to the state of the empodium and true claws on all legs (pad-like) and having four pairs of prodorsal setae on prominent lobes, different from the genera *Tetranychopsis* Canestrini and *Notonychus* Davis (diagnosed with absence of lobes). With the status of *Bryocopsis* species solved, the designation of Meyer’s eight *Bryobia* species remains ambiguous.

The eight *Bryobia* species in Appendix A do not follow the diagnosis (sensu Bolland et al. [5]) of the genus *Bryobia* only in terms of the true claw of leg I morphology. However, in all other morphological aspects, they satisfy the *Bryobia* diagnosis (number of idiosomal setae, number of setae on coxa II, etc). A similar example could be found when Mitrofanov [13] introduced the subgenus *B.* (*Bryobiopsis*) to accommodate those Bryobiine species which lack prodorsal projections/lobes (a character of *Pseudobryobia*, but variably found in *Bryobia*) and have one seta on coxae II (only found in the *Bryobia* species). Contrastingly, the genus *Sinobryobia* Ma et al. was described with a unique set of characteristics such as a spinneret on the palp tarsus present and two pairs of pseudanal setae. The authors placed this genus in its own tribe, Sinobryobiini. However, Bolland et al. [5] placed this genus in the tribe Bryobiini, considering the presence of a spinneret as misidentification and two pairs of pseudanal setae not as a tribe character.

In the current scenario, after discussing all that taxonomically and systematically happens in the subfamily Bryobiinae, we consider that those eight *Bryobia* species (Appendix A) can neither be suitably placed in the tribe Bryobiini nor be assigned to the tribe Hystrichonychini. Those eight species pose an intermediate situation. However, complete morphological analysis could place them close to the tribe Bryobiini. Probably, it may suggest erecting a new subtribe under the tribe Bryobiini. Subtribes have never been added/erected in the superfamily Tetranychoidea. However, this taxonomic rank is used in the other families of Prostigmata (Cheyletidae) and Mesostigmata (Phytoseiidae). We could propose assigning those eight species to a new subtribe under the tribe Bryobiini. A detailed phylogenetic analysis and exhaustive systematics could provide enough confidence to suggest a new subtribe in the superfamily Tetranychoidea.

### 3.10. Key to Subgenera and Species Groups of the Genus Bryobia

1.Solenidion with its associated tactile setae forming a duplex on both leg tarsi III–IV ......................................................................................... **subgenus *Bryobia* Koch** ............ 2

-Solenidion with its associated tactile setae not forming a duplex on either both leg tarsi III–IV or only on leg tarsus IV .................................................................................... 4

2.Setae *f*1 present laterally, along the margin, and the distance *f*1–*f*1 is always greater than *f*2–*f*2 ............................................................................................*praetiosa* species group

-Setae f1 present centrally or sublaterally, not marginally .............................................. 3

3.Setae *f*1 present sublaterally where the distance *f*1–*f*1 is shorter than *f*2–*f*2 .......................................................................................................... *artemisiae* species group

-Setae *f*1 present centrally, aligned with other three pairs of dorsocentral setae ....................................................................................................... *neoephedrae* species group

4.Solenidion with its associated tactile setae not forming a duplex on both leg tarsi III–IV and solenidion present at some distance from its nearest tactile setae on both leg tarsi III–IV ............................................ **subgenus *Allobia* Livschits and Mitrofanov**........... 5

-Solenidion with its associated tactile setae forming a duplex only on leg tarsus III and duplex absent on leg tarsus IV ..... **subgenus *Lyobia* Livschits and Mitrofanov** ...... 7

5.Setae *f*1 present laterally, along the margin, and the distance *f*1–*f*1 is always greater than *f*2–*f*2 ......................................................................................... *pritchardi* species group

-Setae *f*1 not present laterally ............................................................................................... 6

6.Setae *f*1 present centrally, aligned with other dorsocentral setae, and the distance *f*1–*f*1 is always shorter than *f*2–*f*2 ....................................................... *abbatielloi* species group

-Setae *f*1 present sub laterally, neither aligned with other dorsocentral setae nor present marginally, and the distance *f*1–*f*1 could be shorter or longer than *f*2–*f*2 ..............................………………………………………………… *deserticola* species group

7.Setae *f*1 present centrally or subcentrally, usually aligned with other dorsocentral setae, and the distance *f*1–*f*1 always shorter than *f*2–*f*2 ............................. *eurotiae* species group

-Setae *f*1 present laterally or sublaterally ............................................................................ 8

8.Setae *f*1 present sublaterally and the distance *f*1–*f*1 either shorter or longer than *f*2–*f*2 ........................................................................................................ *sarothamni* species group

-Setae *f*1 present laterally and the distance *f*1–*f*1 always longer than *f*2–*f*2 ….................................................................................................... *rubrioculus* species group

## Figures and Tables

**Figure 1 insects-15-00859-f001:**
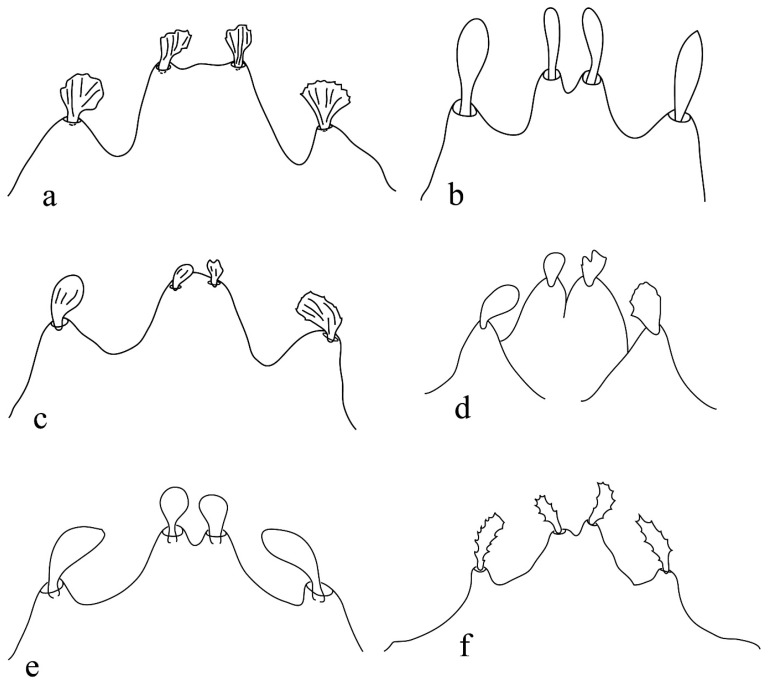
Development and shape of prodorsal idiosomal lobes in some *Bryobia* species redrawn from (**a**) *B*. *coatesi* Meyer [7], (**b**) *B*. *gushariensis* Livschits and Mitrofanov [33], (**c**) *B*. *geyeri* Meyer [7], (**d**) *B*. *pandayi* Eyndhoven and Vacante [34], (**e**) *B*. *convolvulus* Tuttle and Baker [35], and (**f**) *B*. *dianthi* Mitrofanov and Sharonov [36].

**Figure 2 insects-15-00859-f002:**
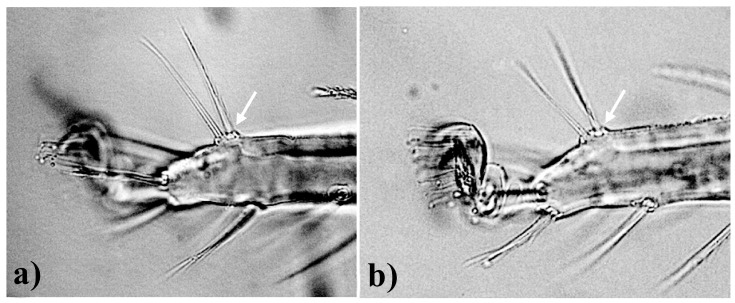
Leg tarsi (**a**) III and (**b**) IV, depicting (by white arrow) the presence of duplex structure by associated setae in *B*. *praetiosa*.

**Figure 3 insects-15-00859-f003:**
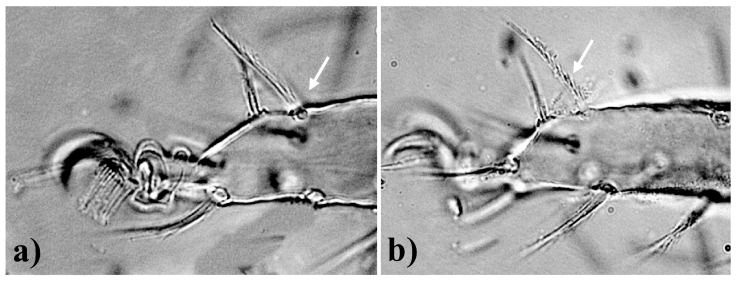
Leg tarsi (**a**) III and (**b**) IV, depicting (by white arrow) the absence of duplex structure by dissociated setae in *B*. *abbatielloi*.

**Figure 4 insects-15-00859-f004:**
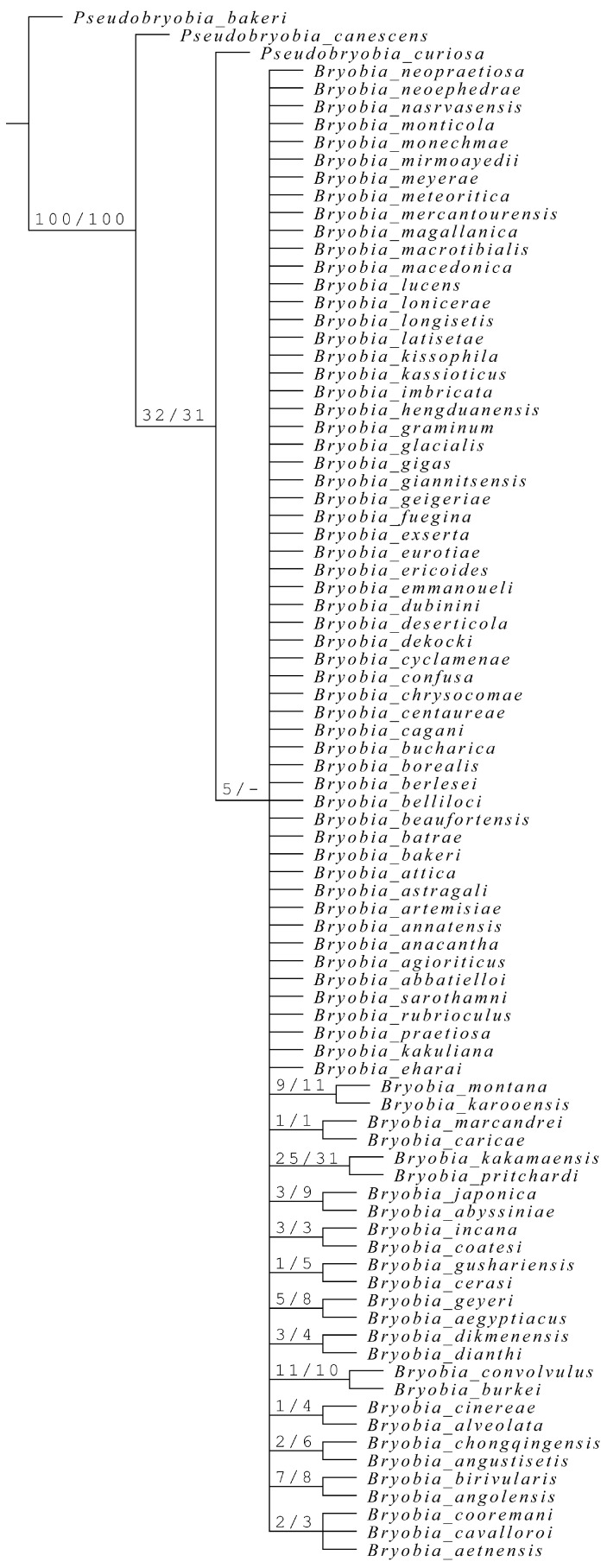
Dendrogram based on morphological characteristics for the assessment of proposed subgenera in the genus *Bryobia* species via TNT, –using new technology search.

**Figure 5 insects-15-00859-f005:**
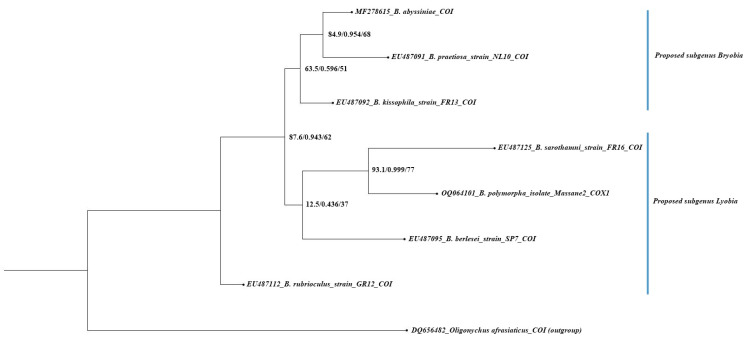
Maximum likelihood tree based on molecular sequences of COI gene, acquired from NCBI database for the assessment of proposed subgeneric assignment of *Bryobia* species (three values at each node represent SH-aLRT support %, aBayes parametric test, and ultrafast bootstrap support %).

**Figure 6 insects-15-00859-f006:**
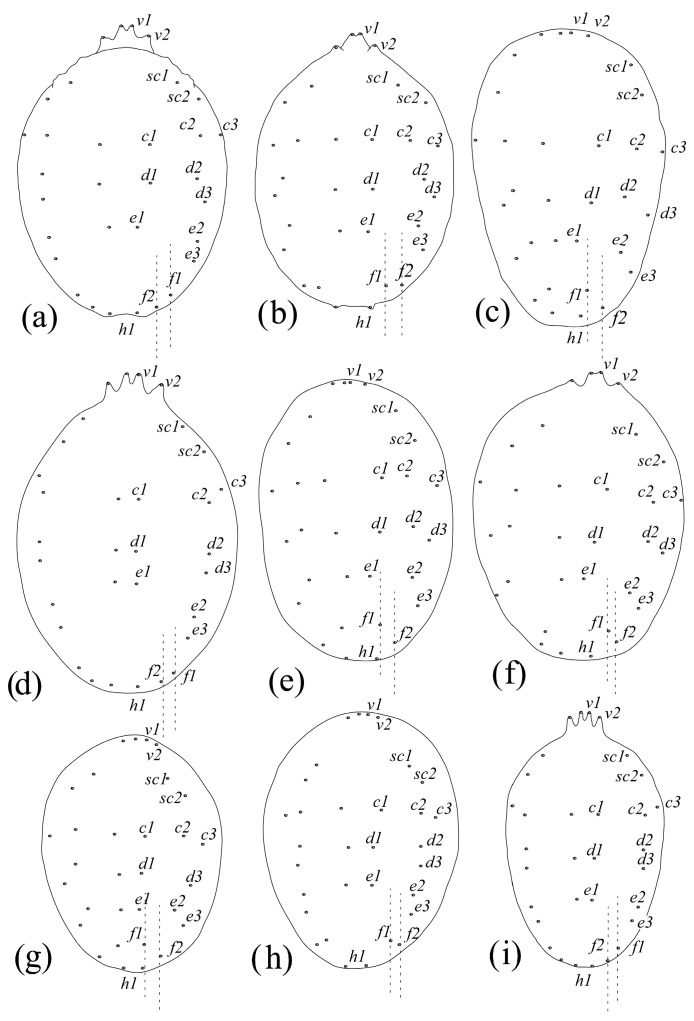
Position of setae *f*1 and relative distance of *f*1–*f*1 vs. *f*2–*f*2 (shown by dashed line) in the exemplary species of each species group: (**a**) *Bryobia* (*B*.) *praetiosa* (redrawn from Pritchard and Baker [2]), (**b**) *B*. (*B*.) *artemisiae* (redrawn from Livschits and Mitrofanov [12]), (**c**) *B*. (*B*.) *neoephedrae* (redrawn from Strunkova and Mitrofanov [37]), (**d**) *B*. (*A*.) *pritchardi* (redrawn from [38]), (**e**) *B*. (*A*.) *abbatielloi* (redrawn from Smiley and Baker [31]), (**f**) *B*. (*A*.) *deserticola* (redrawn from Meyer and Ueckermann [39]), (**g**) *B*. (*L*.) *eurotiae* (redrawn from Meyer and Ueckermann [39]), (**h**) *B*. (*L*.) *sarothamni* (redrawn from Meyer and Ueckermann [39]), and (**i**) *B*. (*L*.) *rubrioculus* (redrawn from Meyer and Ueckermann [39]).

## Data Availability

All necessary data are provided in this paper.

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
