# Peer review of "Taxonomy of the Genus Bryobia Koch (Acari: Tetranychidae): Reconsideration of Subgenera and Updated Species Groups"

_insects, 2024, doi:10.3390/insects15110859_

Round 1

Reviewer 1 Report

Comments and Suggestions for Authors

This paper proposes an updated classification of Bryobia species into subgenera and species groups, essentially based on the literature. Their classification can facililitate species-level identification by having infrageneric groupings that are defined with two main characters, the presence of duplex setae on tarsi III-IV and the position of f1-f2 setae. Despite weaknesses (see below), this paper can be useful/valuable for future research on the genus.  It also provides some useful discussion on subgeneric concepts and related genera by previous workers.

I could not access the Supplem. Tables (the link did not work). I think the authors should always be asked by journals to submit supplem. material with the main file (without the need to click on additional links). But I have a good idea of what’s included in these tables.

Main weaknesses of the paper are:

 (1) that the updated classification is primarily literature-based, without any substantial support/analysis based on specimen observations. It is not bad per se, but it is of more limited value. That your analysis is largely ‘literature-based’ should be mentioned in your abstract.

 (2) the subgeneric classification is essentially based on a single attribute (duplex setae; or two attributes, one for tarsus III, another for tarsus IV), and the species group on another attribute (relative positions of f1-f2). It is OK, but a single character is quite limited, especially if its actual phylogenetic stability is poorly know. Indeed:

 (3) These 2  characters are of uncertain phylogenetic significance, probably more so for the position of f1-f2. The proposed classification implies that the positions of f1-f2 evolved twice (in each subgenus), suggesting that their positions are plastic (not always stable). It’s ok, as we have to work with what we have. But an analysis based on so few features cannot test much the significance of the features. Similarly, the morphology-based dendrogram is of limited significance because it (seems) based on very few characters. So, while the paper provides a useful revised scheme, it remains more practical than absolutely reflecting phylogenetic relationship. It is OK, but this should be more clearly mentioned in the paper, by stating that the phylogenetic significance of these features are uncertain, and that this classification is based on their hypothesis on the evolution of characters (and of some previous workers). Personally, despite what I just said about their uncertainty, I agree with the authors that these features (duplex setae, and f1-f2 position) must have SOME phylogenetic significance and are important features to help define (tentatively monophyletic) taxa. These features also have been used valuably in other groups (e.g. Tetranychinae).

(4) the COI-based tree is very limited in magnitude and all made from sequences of GenBank, not theirs.  It also has issues related to what subgenus each species belong (see below). more than once in the paper I was confused about whether they were talking about the own proposed subgeneric classification, or previous classification(s).

Dendrogram (Fig 4):

-In method and caption should also probably give the number of characters used. If you have 0 shared characters between Pseudobryobia and Bryobia, it may be because you used a very small number of features, which makes it weak.

-In caption: when you say “subgenus Allobia” etc., is it YOUR concept that you use (or previous concepts)? If so, say “herein assigned to…” or “assigned to Allobia in the classification followed herein”.  But, since subgenera are broken in different clades in this dendrogram, you acutally used PREVIOUS subgenera? Need clarification.

-I think it’d be useful to have species names on the graph instead of numbers. Numbers are very impractical to interpret figure.

Molecular-based tree:

The tree has only 7 sequences and all were taken from GenBank, as I understand, none produced by the authors. If these sequences are to be used in this paper, then they should give details on who made the morphology-based identifications of the voucher specimens, and how much it is reliable.  The caption should clearly state that sequences are from GenBank.

-Your Method and caption should state that maximum likelihood was used, for clarity.

-I find the tree confusing because:

-they show a ‘proposed subgenus Lyobia”, including 4 species, including 2 species (berlesei and rubrioculus) that they elsewhere include in subgenus Allobia.

-this group of 4 species for ‘Lyobia’ is paraphyletic!  That should be mentioned in the text, and considered when defining the subgenus. 

Introduction:

Lines 67-68: maybe briefly mention which characters are overlapping? 

Lines 80-81: confusing sentence.

Results

-lines 133-134: ambiguous. This suggests that their (original) illustrations represent distinct species, which is probably not what you wanted to imply.  Isn’t rather that Koch’s illustrations are too general or vague, and therefore insufficient for species identifications?

135-136:  are we here talking only about spp that were synonymized under B praetosia? Specify/adjust

-lines 165-166: not clear what is meant here. “exist”? 

-166-167: these species groups are not ambiguous because authors did not follow them.

-lines 172-173: I doubt that the “widely accepted classification of Bolland et al.” is largely based on previous work, such as Gutierrez 1985 (1.1.4 Systematics chapter of Spider Mites book), so I’d suggest to also cite that reference at least, even if Gutierrez is an author of Bolland et al. 1998.

174-175: it doesn’t make the identity of Meyer’s species doubtful. But their taxonomic position may need to be verified indeed (it’s not the same thing).

-285-186: not clear. Do you mean it varied intraspecifically?

-line 261: you defined presence vs absence of duplex in your Method, so you don’t need to always add ‘associated’ when you say duplex.

-lines 263-264: it’s not quite a ‘result’. That this character is clearly binary, with 2 discrete states, does not mean that they are necessary apomorphic, i.e. present in the ancestor of a subgenus. You simply consider that it may be so. Perhaps replace “As a result” by “Based on this” to resolve this.

-lines 337-339: you cannot really draw these conclusions; it’s a bit like circular reasoning. And for ‘species delineation’: it has little to do with your tree and subgeneric division, so I suggest you delete this from there.

-342: I would just call this a ‘molecular-based tree’ or a ‘COI-based tree’.

-lines 451-458:  here you synonymize 3 subgenera under Allobia. It seems warranted to reiterate the defining characters of the 3 subgenera that you are synonymizing, and why these characters fit with the (senior) subgenus Allobia. For instance, Bryobiopsis was erected by Mitrofanov for spp. lackign prodorsal lobes. Perhaps explain that the lobes can be present or absent in the newly defined subgenus Allobia (even if that may be the case in other subgenera).

-You synonymized Lyobia under Allobia, but alter in section 3.7 you (re)define subgenus Lyobia. There is a confusing problem here.

-lines 593-599: clarify in the text that you could not assign these 5 spp. “to any subgenus”. However, perhaps you should specify whether they were assigned to a given subgenus in the past and why you think that they cannot be kept in such subgenus.

-line 610: Because you had just explained that Arabuli et al considered that Smiley and Baker inversed mistakenly the diagnosis of the 2 genera, you should rephrase this to avoid confusion, to something like “…following the diagnosis of the genus Bryobia sensu Smiley and Baker, and therefore should be classified in the genus Pseudobryobia, based on coxa II setation”

-lines 638-639: Mention Gutierrez (1985) there because he had already placed Bryocopsis in Hysteronychini, before Bolland et al. 1998.

Comments on the Quality of English Language

Although for non-native English speakers this paper is fairly well written, the English needs improvement (see examples in the PDF for where it needs improvement in clarity).

Author Response

Author’s response to the comments of the Reviewer 1

Comment#1 The updated classification is primarily literature-based, without any substantial support/analysis based on specimen observations. It is not bad per se, but it is of more limited value. That your analysis is largely ‘literature-based’ should be mentioned in your abstract.

Response: The abstract has been updated with necessary changes in the revised version of the manuscript.

Comment#2 The subgeneric classification is essentially based on a single attribute (duplex setae; or two attributes, one for tarsus III, another for tarsus IV), and the species group on another attribute (relative positions of f1-f2). It is OK, but a single character is quite limited, especially if its actual phylogenetic stability is poorly known.

Response: The persistence of diagnostic morphological character is significantly important at taxonomic ranks, rather than the number of characters. In the recent review of the genus Pseudobryobia (Arabuli et al. 2019), which is morphologically close to the genus Bryobia, the updated diagnosis of both genera were provided based on combination of characters. However, the precise analysis of those characters show that these two genera could only be differentiated based on a single persistant character i.e. leg coxal setal formula (2-2-1-1 in Pseudobryobia and 2-1-1-1 in Bryobia). Arabuli et al. (2019) also provided excellent debate upon persistence of this character which is both literature based and observation of different specimens.

Similarly, the subgenera erected in the genus Bryobia by Mitrofanov and Strunkova (1971) and Mitrofanov (1973) were based on combination of morphological characters. However, it was important to understand the significance of those characters as many species were described since then. In the presented results, the detailed analysis of all characters revealed that state of duplex setae on leg tarsi III-IV is consistently present and persistently described in most Bryobia species. Furthermore, the proposed subgeneric divisions were supported by adequate number of species and no variations were found. The actual phylogenetic stability of the subgeneric characters are explained in the updated analysis. The detailed integrated taxonomic studies may further enhance our understanding of different taxa in the tribe Bryobiini.

Comment# 3 These 2 characters are of uncertain phylogenetic significance, probably more so for the position of f1-f2. The proposed classification implies that the positions of f1-f2 evolved twice (in each subgenus), suggesting that their positions are plastic (not always stable). It’s ok, as we have to work with what we have. But an analysis based on so few features cannot test much the significance of the features. Similarly, the morphology-based dendrogram is of limited significance because it (seems) based on very few characters. So, while the paper provides a useful revised scheme, it remains more practical than absolutely reflecting phylogenetic relationship. It is OK, but this should be more clearly mentioned in the paper, by stating that the phylogenetic significance of these features are uncertain, and that this classification is based on their hypothesis on the evolution of characters (and of some previous workers). Personally, despite what I just said about their uncertainty, I agree with the authors that these features (duplex setae, and f1-f2 position) must have SOME phylogenetic significance and are important features to help define (tentatively monophyletic) taxa. These features also have been used valuably in other groups (e.g. Tetranychinae).

Response: The different species groups, added in each proposed subgenus in the present study, are not taxonomic ranks and would not have any phylogenetic significance. The position of fourth pair of dorscentral setae (f1) has been used for generic differentiation in the tribe Hystrichonychini. In the genus Bryobia, the position of setae f1 was also previously used in combination for the subgeneric differentiation by Mitrofanov and Strunkova (1971) and Mitrofanov (1973). However, in the present manuscript, the arguments have been provided that these characters could not support the assignment of species in any subgenus sensu Mitrofanov and Strunkova (1971) and Mitrofanov (1973). For the time, this character is helpful for species classification in each proposed subgenus. We agree with the reviewer suggestions to make changes in the phylogenetic results and the necessary changes have been made in the revised version of the manuscript.

Comment# 4 the COI-based tree is very limited in magnitude and all made from sequences of GenBank, not theirs.  It also has issues related to what subgenus each species belong (see below). more than once in the paper I was confused about whether they were talking about the own proposed subgeneric classification, or previous classification(s).

Response: The necessary changes have been made in the revised version of the manuscript. It has been made clear that the sequences used in the present study, were entirely acquired from the NCBI GenBank database.

Dendrogram (Fig 4):

Comment# 4 In method and caption should also probably give the number of characters used. If you have 0 shared characters between Pseudobryobia and Bryobia, it may be because you used a very small number of features, which makes it weak.

Response: The phylogenetic analysis of the proposed subgenera in the genus Bryobia in the present study, includes the morphological characters which were distinctly used by Mitrofanov and Strunkova (1971), Mitrofanov (1973) for subgenera and in the update generic diagnosis by Arabuli et al. (2019). The only character which is not shared between the genera Pseudobryobia and Bryobia is the number of setae on coxae II (Arabuli et al. 2019). All other characters are shared between both genera in their different states. Hence, the morphological character being distinct is of significance rather than the number of characters.

            Furthermore, other morphological characters were used for species differentiation in each genus. If such characters were to use at generic differentiation, it may reduce the significance of persistant character and may dilute the diagnosis of the taxonomic rank. The perfect example in the current scenario, is the diagnosis of the genera Pseudobryobia and Bryobia. Previously, these genera were differentiated based on absence and presence of propodosomal lobes, respectively. However, this character was found variably present in the Bryobia species and number of setae on coxae II was signified for generic diagnosis (Arabuli et al. 2019).

Comment# 5 In caption: when you say “subgenus Allobia” etc., is it YOUR concept that you use (or previous concepts)? If so, say “herein assigned to…” or “assigned to Allobia in the classification followed herein”.  But, since subgenera are broken in different clades in this dendrogram, you acutally used PREVIOUS subgenera? Need clarification.

Response: This confusion has been corrected in whole revised manuscript.

Comment# 6 I think it’d be useful to have species names on the graph instead of numbers. Numbers are very impractical to interpret figure.

Response: The dendrogram has been updated based on new analysis. Additionally, the name of species in the previous dendrogram were not added as it made the tree impossible to read.

Molecular-based tree:

Comment# 7 The tree has only 7 sequences and all were taken from GenBank, as I understand, none produced by the authors. If these sequences are to be used in this paper, then they should give details on who made the morphology-based identifications of the voucher specimens, and how much it is reliable.  The caption should clearly state that sequences are from GenBank.

Response: It is clear in Methodology and citation to each acquired sequence were provided. The sequences were from published works and the publishing authors mentioned in their methodology that they performed morphometric analysis. The figure captions have been updated.

Comment# 8 Your Method and caption should state that maximum likelihood was used, for clarity.

Response: The figure captions have been updated.

Comment# 9 I find the tree confusing because:

-they show a ‘proposed subgenus Lyobia”, including 4 species, including 2 species (berlesei and rubrioculus) that they elsewhere include in subgenus Allobia.

-this group of 4 species for ‘Lyobia’ is paraphyletic!  That should be mentioned in the text, and considered when defining the subgenus. 

Response: The results have been updated in the revised version of the manuscript.

Introduction:

Lines 67-68: maybe briefly mention which characters are overlapping? 

Response: These characters have been discussed in detail in the results and discussion. Also, this statement has been modified in the revised version, as suggested.

Lines 80-81: confusing sentence.

Response: The statement has been rephrased.

Results

lines 133-134: ambiguous. This suggests that their (original) illustrations represent distinct species, which is probably not what you wanted to imply.  Isn’t rather that Koch’s illustrations are too general or vague, and therefore insufficient for species identifications?

Response: It could be true as suggested by the reviewer about Koch’s illustrations. However, this statement is the part of taxonomic history and may explain the difficult taxonomic status of the genus Bryobia. The statement is not the result of the present study.

135-136:  are we here talking only about spp that were synonymized under B praetosia? Specify/adjust

Response: The statement has been explained in the revised manuscript.

lines 165-166: not clear what is meant here. “exist”? 

Response: The statement has been rephrased in the revised version.

166-167: these species groups are not ambiguous because authors did not follow them.

Response: The statement has been rephrased in the revised version.

lines 172-173: I doubt that the “widely accepted classification of Bolland et al.” is largely based on previous work, such as Gutierrez 1985 (1.1.4 Systematics chapter of Spider Mites book), so I’d suggest to also cite that reference at least, even if Gutierrez is an author of Bolland et al. 1998.

Response: The citation has been added in the revised manuscript.

174-175: it doesn’t make the identity of Meyer’s species doubtful. But their taxonomic position may need to be verified indeed (it’s not the same thing).

Response: The statement has been updated in the revised manuscript.

-285-186: not clear. Do you mean it varied intraspecifically?

Response: The statement does not mean that the character varied interspecifically. The statement has been updated in the revised version.

line 261: you defined presence vs absence of duplex in your Method, so you don’t need to always add ‘associated’ when you say duplex.

Response: The necessary changes have been made in the revised manuscript.

lines 263-264: it’s not quite a ‘result’. That this character is clearly binary, with 2 discrete states, does not mean that they are necessary apomorphic, i.e. present in the ancestor of a subgenus. You simply consider that it may be so. Perhaps replace “As a result” by “Based on this” to resolve this.

Response: The statement has been corrected in the revised manuscript.

lines 337-339: you cannot really draw these conclusions; it’s a bit like circular reasoning. And for ‘species delineation’: it has little to do with your tree and subgeneric division, so I suggest you delete this from there.

Response: The statement has been deleted.

342: I would just call this a ‘molecular-based tree’ or a ‘COI-based tree’.

Response: The statement has been corrected in the revised manuscript.

lines 451-458:  here you synonymize 3 subgenera under Allobia. It seems warranted to reiterate the defining characters of the 3 subgenera that you are synonymizing, and why these characters fit with the (senior) subgenus Allobia. For instance, Bryobiopsis was erected by Mitrofanov for spp. lackign prodorsal lobes. Perhaps explain that the lobes can be present or absent in the newly defined subgenus Allobia (even if that may be the case in other subgenera).

Response: The necessary changes have been made in the revised manuscript.

You synonymized Lyobia under Allobia, but alter in section 3.7 you (re)define subgenus Lyobia. There is a confusing problem here.

Response: The statement has been corrected in the revised manuscript.

lines 593-599: clarify in the text that you could not assign these 5 spp. “to any subgenus”. However, perhaps you should specify whether they were assigned to a given subgenus in the past and why you think that they cannot be kept in such subgenus.

Response: This information was already provided in the Table S2.

line 610: Because you had just explained that Arabuli et al considered that Smiley and Baker inversed mistakenly the diagnosis of the 2 genera, you should rephrase this to avoid confusion, to something like “…following the diagnosis of the genus Bryobia sensu Smiley and Baker, and therefore should be classified in the genus Pseudobryobia, based on coxa II setation”

Response: The statement has been corrected in the revised manuscript.

lines 638-639: Mention Gutierrez (1985) there because he had already placed Bryocopsis in Hysteronychini, before Bolland et al. 1998.

Response: The citation has been added in the revised manuscript.

Reviewer 2 Report

Comments and Suggestions for Authors

In the reviewed MS a group of authors from Saudi Arabia guided by a very well-known acarologist from Brazil propose a new system of the genus Bryobia, a taxon of plant-feeding spider mites which includes many economically important pests of crops. The authors analyzed current morphological data on the genus Bryobia, revised old classification of this genus, proposed new synonymy, and established several new groups of species. The authors also attempted to analyze the data from GenBank (COI sequences of Bryobia) and compare their results with morphoilogical data. This part of the study is the weakest part of the MS. The molecular analysis has been performed incorrectly. The authors did not use any out-groups and presented NJ tree instead of ML or Bayesian tree, which is current standard for mol.phyl. studies. Additionally, the authors used only 7 sequences from GB, although currently there are about 150 COI sequences. The authors show the results of their cladistics analysis as a dendrogram, and this dendrogram conflicts with the new groups of species, which they established. This is very confusing. The MS is very wordy, it could be shorter. Some repetitions could be removed from Results, especially those sentences repeating the Introduction. This MS needs serious revision; the analyses should be redone and the results need to be presented in a different form. Some additional remarks are below.

2 Probably ; in the title should be changed to :

16 based on absence and/or presence of duplex setae – please, simplify this, it is hard to read

25 viz; subgenus – change ; to : check this in the entire text

89 critically reviewed to confirm – please, explain better the methodology

93 each morphological character was evaluated for its suitability – please, explain better the methodology, how exactly the evaluation was done

96 and tactile on – a word was missed

99 Cladistic analysis – it looks like you did not perform classical cladistics analysis of the morphological traits. Please, explain why you did not do this and preferred hierarchical method? This looks inadequate

111,112 were analysed by multivariate analysis with observational analysis using Pearson distance with complete linkage – please, explain the choice of the method. Please give a brief description of this method and what exactly did you expect to evaluate using this method, which hypothesis you tested and which results exactly you obtained.

122-125 please, perform Maximum Likelihood analysis with all sequences of Bryobia and out groups from GenBank. The molecular phylogenetic analysis was performed and inadequately

145 generation type – please revise this word combination

158 Table S1 concludes – may be “summarizes” is better?

187 table S2 – Table, check in the entire text

203 dorsal lobes – please provide drawings of all morphological structures, which you mention in the text, and indicate them with arrows in the drawings.

274 coxal formula 2-1-1-1 – please explain this formula better, it may be unclear for an unprepared reader

275 dorsal propodosoma with lobes well developed (outer lobes separated from inner lobes with deep incision), lobes poorly developed or absent – this sounds confusing, please revise this

281 duplex present – this is plesiomorphy. Why do you establish a new group based on plesiomorphic trait? Please, explain this, otherwise this is confusing

293 specially - revise this word

297 if both the setae; solenidion and the tactile, are placed – not clear, please, revise

302, 305 Figure 2,3. Leg tarsi a)III and b) IV – please revise, e.g. Leg tarsi III (a) and IV (b). Please, indicate all setae with arrows and white labels. Please provide microphotographs of the species form all new groups and unambiguously mention this in the captions

Fig.4 red lines – these lines are brown on my display. Please, revise colors

Fig.5 please, redo the mol phyl analysis

Sections 3.4.1. and 3.4.2. – these two sections need serious revision. Interpretation of COI analysis in inadequate.

346 According to M & M you performed NJ analysis, why do mention ML analysis here?

352,353 This was reported due to infection of Wolbachia parasite in Bryobia species [20]. The Wolbachia – why do you write about Wlobachia gere? This is completely out of the scope of this MS.

367 Exempler – please, revise this word

3.9. Notes on the Bryobia species described by Meyer 1974, 1987 – please, check if the reference follows the format of the journal

658 comfortably – revise this word

660 It probably suggests to erect -  please, revise this

663-666 please explain this better and give some stronger support or remove this speculation from the text

Comments on the Quality of English Language

minor

Author Response

Author’s response to the comments of the Reviewer 2

Comment#1 The molecular analysis has been performed incorrectly. The authors did not use any out-groups and presented NJ tree instead of ML or Bayesian tree, which is current standard for mol.phyl. studies.

Respone: The molecular analysis has been re-done in the revised version of the manuscript. The methodology and result sections have been updated. Based on morphology, the Pseudobryobia genus is the closest to the genus Bryobia. However, no gene sequence of any Pseudobryobia species is present on the GenBank database. Hence, no outgroup was could be suitable considered. But as reviewer suggested, Oligonychus afrasiaticus, a species from other subfamily has been added in the analysis.

Comment#2 Additionally, the authors used only 7 sequences from GB, although currently there are about 150 COI sequences.

Response: The available COI sequences submitted to GenBank database, are indeed more than seven asit was mentioned in the results and discussion section. The reason to use only seven sequences was, only those were morphology identified till species level. Using other sequences of unknown morphology would lead the results away from the main objective of the study.

Comment#3 The authors show the results of their cladistics analysis as a dendrogram, and this dendrogram conflicts with the new groups of species, which they established. This is very confusing.

Response: The group of species or species groups proposed in each proposed subgenus in the present study, are not taxonomic ranks. Hence, their significance is to ease the identification where the number of species are more. They do not necessarily have the phylogenetic relationship. The subgeneric divisions proposed in the present study, however, do show supporting clades. Still, the phylogenetic significance of those characters used for subgeneric designation is not completely understood, but their diagnostic significance has been well studied and established in the subfamily Bryobiinae. Additionally, the results for both morphological and molecular analysis have been been updated in the revised version.

Comment#4 The MS is very wordy, it could be shorter. Some repetitions could be removed from Results, especially those sentences repeating the Introduction.

Response: The necessary changes have been made in the revised version of the manuscript, where some statements are omitted while others are rephrased.

Comment#5 Probably ; in the title should be changed to :

Response: It has been changed in the entire manuscript.

Comment#6 based on absence and/or presence of duplex setae – please, simplify this, it is hard to read

Response: The statement has been simplified in the revised version.

Comment#7 viz; subgenus – change ; to : check this in the entire text

Response: It has been changed in the entire manuscript.

Comment#8 critically reviewed to confirm – please, explain better the methodology

Response: The statement has been explained in the revised manuscript.

Comment#9 each morphological character was evaluated for its suitability – please, explain better the methodology, how exactly the evaluation was done

Response: The statement has been explained in the revised manuscript.

Comment#10 and tactile on – a word was missed

Response: The word “setae” was missing and has been added in the revised version of the manuscript.

Comment#11 Cladistic analysis – it looks like you did not perform classical cladistics analysis of the morphological traits. Please, explain why you did not do this and preferred hierarchical method? This looks inadequate

Response: The reason to use Heirarchical analysis was to expand our understanding about the phylogenetic relationship of the newly proposed subgenera in the genus Bryobia. The method included clustering the taxa based on similarities in the morphological traits and group them in their position of hierarchy (taxonomic divisions). It could be one of the reasons that the species groups do not follow this divisions as they are not certain taxonomic ranks. There may be different school of thoughts when it comes phylogenetic studies, however, the cladistic analysis is considered as analysis of hierarchical branching diagram (Nelson 1979).

Comment#12 111,112 were analysed by multivariate analysis with observational analysis using Pearson distance with complete linkage – please, explain the choice of the method. Please give a brief description of this method and what exactly did you expect to evaluate using this method, which hypothesis you tested and which results exactly you obtained.

Response: The main hypothesis to test was whether diagnostic characters used to erect existing six subgenera of the genus Bryobia hold significance to be used for taxonomic divisions. The expectations were to evaluate the significance and persistant nature of morphological traits used as differentiating or diagnostic and certain taxonomic ranks. A detailed literature based debate has already been provided in the manuscript. This multivariate analysis was performed to support the hypothesis that the six subgeneric divisions in the genus Bryobia are based on unstable characters. The multivariate analysis hold many different techniques among them is the clustering of observations based on similarities of the characters (dataset). Hence it was suitably used.

Comment#13 122-125 please, perform Maximum Likelihood analysis with all sequences of Bryobia and out groups from GenBank. The molecular phylogenetic analysis was performed and inadequately

Response: The anaylsis has been re-done and the results have been updated.

Comment#14 145 generation type – please revise this word combination

Response: The statement has been revised.

Comment#15 158 Table S1 concludes – may be “summarizes” is better?

Response: The word has been replaced.

Comment#16 187 table S2 – Table, check in the entire text

Response: Necessary changes have been made in the entire text.

Comment#17 203 dorsal lobes – please provide drawings of all morphological structures, which you mention in the text, and indicate them with arrows in the drawings.

Response: The character of propodosomal lobes was already illustrated precisely in the manuscript. Other features used in the whole manuscript, have also been illustrated and correctly cited.

Comment#18 274 coxal formula 2-1-1-1 – please explain this formula better, it may be unclear for an unprepared reader

Response: This is the standard way of writing coxal formula, specially in the family Tetranychidae and generally in most groups of the suborder Prostigmata.

Comment#19 275 dorsal propodosoma with lobes well developed (outer lobes separated from inner lobes with deep incision), lobes poorly developed or absent – this sounds confusing, please revise this

Response: the necessary changes have been made in the revised manuscript.

Comment#20 281 duplex present – this is plesiomorphy. Why do you establish a new group based on plesiomorphic trait? Please, explain this, otherwise this is confusing

Response: The phylogenetic significance of this character needs further understanding. This character state has been significantly used to differentiate among the tribes (above genus level) of the subfamily Tetranychinae. In the present study, we believe that it is not a plesiomorphic trait because the character of duplex setae has not been used to differentiate among tribes and genera of the subfamily Bryobiinae. It has different states in the species of the genus Bryobia which is the basis for the subgeneric divisions. The previous analysis was inadequate as mentioned by both respected reviewers and the authors found out that the species added in the analysis made the results far away from the main hypothesis and objective. Hence, the updated results are provided.

Comment#21 293 specially - revise this word

Response: The word has been revised.

Comment#22 297 if both the setae; solenidion and the tactile, are placed – not clear, please, revise

Response: The statement has been clarified.

Comment#23 302, 305 Figure 2,3. Leg tarsi a)III and b) IV – please revise, e.g. Leg tarsi III (a) and IV (b). Please, indicate all setae with arrows and white labels. Please provide microphotographs of the species form all new groups and unambiguously mention this in the captions

Response: The figure captions have been revised. The setae have been indicated with white arrows. The microphotograph of all the forms of the species groups are not possible to be provided. However, they were illustrated clearly in the Figure 6.

Comment#24 Fig.4 red lines – these lines are brown on my display. Please, revise colors

Response: The updated figure does not have any color and labels are clearly presented.

Comment#25 Fig.5 please, redo the mol phyl analysis

Response: The molecular analysis has been updated.

Comment#26 Sections 3.4.1. and 3.4.2. – these two sections need serious revision. Interpretation of COI analysis in inadequate.

Response: The results have been updated in the revised version of the manuscript.

Comment#27 346 According to M & M you performed NJ analysis, why do mention ML analysis here?

Response: The corrections have been made in the text.

Comment#28 352,353 This was reported due to infection of Wolbachia parasite in Bryobia species [20]. The Wolbachia – why do you write about Wlobachia gere? This is completely out of the scope of this MS.

Response: The mention of Wolbachia is just to reference its role in the evolution of the mitochondrial genome. It was referenced towards the results acheived through molecular anaylysis.

Comment#29 367 Exempler – please, revise this word

Response: The word has been revised.

Comment#30 3.9. Notes on the Bryobia species described by Meyer 1974, 1987 – please, check if the reference follows the format of the journal

Response: It has been corrected in the whole revised manuscript.

Comment#31 658 comfortably – revise this word

Response: The word has been revised in the manuscript.

Comment#32 660 It probably suggests to erect -  please, revise this

Response: The statement has been corrected in the revised manuscript.

Comment#33 663-666 please explain this better and give some stronger support or remove this speculation from the text

Response: The statement mentioned is “The addition of subtribes has never been practiced in the superfamily Tetranychoidea.” It is well supported based on the existing systematics and taxonomy of the superfamily and is widely accepted by the acarologists. It would not be correct to remove this statement, hence it is retained. The statement is modified to simplify it.

Round 2

Reviewer 2 Report

Comments and Suggestions for Authors

30-33 The phylogenetic analysis of these proposed subgenera was also attempted These three subgenera are differentiated based on the absence and/or presence of duplex setae on leg tarsus III and IV and supported by multivariate analysis based on morphological characters and available COI sequence on the GenBank database

COMMENT. Here you say what you have done but actually in the Abstract we want to see what was achieved. Therefore, please, say directly in the Abstract (and in the MS) that molecular and morphological analyses did not support monophyly of the proposed subgenera.

Section 2.2. In this section, please explain the choice of statistical method (like you did in comment 11). This is important because you prefer old method and this should be justified.

125-137 In this section, please include data on the outgroup used in the Analysis [e.g. Taxon A was used as out-group, + explanation.]

130-131 The morphological identity of these sequences was confirmed by the submitting authors.

COMMENT. This is not clear. Please explain how you did that. Did you check the vouchers or you are the author of these sequences and the sequences came from your previous study (than give a reference)?

Figure 4-new vs Figure 4-old. In the old version of the Fig. 4 the mite taxa from different subgenera did not form distinct groups corresponding to the different subgenera. However, in the new version of the Fig. 4 you show distinct clusters. According to the modified version of the MS you did NOT reanalyze the morphological dataset. Please, explain why in the revised MS you give a new Figure4 showing new result contradicting your previous Figure 4? Without explanation, it looks like data manipulating.

374 The COI based evolutionary tree constructed based on available sequences of Bryobia species which provides some supports the species to the taxonomic grouping in dendrogram based on morphological characters (Figure 4), as well as the subgeneric divisions…

COMMENT. No, the tree DOES NOT provide any support, it is poorly resolved. Please, perform statistical tests of monophyly of branches (e.g. SH, Alrt) and conclude based on these statistical tests. Please, say directly that the new proposed subgenera are not confirmed either by your molecular and morphological (=Fig. 4 old) analyses.

Section 3.4.2. This section is still written inadequately. Mentioning Wolbachia is speculation here.

SUPPLEMENT. There are no supplementary files in the web system of the journal. Have you submitted them?

Comments on the Quality of English Language

minor to moderate

Author Response

Revision Round II

Author’s response to the comments of the Reviewer 2

Comment#1 30-33 The phylogenetic analysis of these proposed subgenera was also attempted These three subgenera are differentiated based on the absence and/or presence of duplex setae on leg tarsus III and IV and supported by multivariate analysis based on morphological characters and available COI sequence on the GenBank database

Here you say what you have done but actually in the Abstract we want to see what was achieved. Therefore, please, say directly in the Abstract (and in the MS) that molecular and morphological analyses did not support monophyly of the proposed subgenera.

Response: The necessary changes have been made in the Abstract and throught the revised manuscript based on the updated results after data reanalysis.

Comment#2 Section 2.2. In this section, please explain the choice of statistical method (like you did in comment 11). This is important because you prefer old method and this should be justified.

125-137 In this section, please include data on the outgroup used in the Analysis [e.g. Taxon A was used as out-group, + explanation.]

Response: A detailed methodology of the data reanalysis has been provided in the Section 2.2. Also, the outgroups have been clearly mentioned.

Comment#3 130-131 The morphological identity of these sequences was confirmed by the submitting authors.

This is not clear. Please explain how you did that. Did you check the vouchers or you are the author of these sequences and the sequences came from your previous study (than give a reference)?

Response: The phrase “submitting author” referred to those authors who submitted the sequences in the NCBI GenBank database. For this purpose, the appropriate references were cited for each accession number. The statement has been further clarified in the revised version.

Comment# 4 Figure 4-new vs Figure 4-old. In the old version of the Fig. 4 the mite taxa from different subgenera did not form distinct groups corresponding to the different subgenera. However, in the new version of the Fig. 4 you show distinct clusters. According to the modified version of the MS you did NOT reanalyze the morphological dataset. Please, explain why in the revised MS you give a new Figure4 showing new result contradicting your previous Figure 4? Without explanation, it looks like data manipulating.

Response: The Figure 4-old included morphological analysis of most species of the Bryobia genus and three species of the closely related genus Pseudobryobia as outgroup. The Figure 4-new included only the type species of the proposed subgeneric divisions and that of an outgroup. It was performed by adding more morphological characters, which could be found in the descriptions and illustrations of the species and have no interspecific variable states (Please see Table S3). The reason to reduce the number of species was to perform the analysis for the proposed subgeneric divisions by including only type species of each proposed division. The species as present in the new figure, follow same clade distribution as in old figure.

            However, in the revised version of the manuscript, updated analysis method has been used to analyze morphological characters in most species of the genus Bryobia. The result section is also updated in the revised version.

Comment#5 374 The COI based evolutionary tree constructed based on available sequences of Bryobia species which provides some supports the species to the taxonomic grouping in dendrogram based on morphological characters (Figure 4), as well as the subgeneric divisions…

No, the tree DOES NOT provide any support, it is poorly resolved. Please, perform statistical tests of monophyly of branches (e.g. SH, Alrt) and conclude based on these statistical tests. Please, say directly that the new proposed subgenera are not confirmed either by your molecular and morphological (=Fig. 4 old) analyses.

Response: The updated results have been added to the revised version of the manuscript. The materials and method section has been updated, too.

Comment#6 Section 3.4.2. This section is still written inadequately. Mentioning Wolbachia is speculation here.

Response: Section 3.4.2 has been carefully revised.

Comment#7 SUPPLEMENT. There are no supplementary files in the web system of the journal. Have you submitted them?

Response: The supplementary files were added before. This time, a complete zip folder is attached including the revised version of the manuscript and all supplementary tables.

Round 3

Reviewer 2 Report

Comments and Suggestions for Authors

In the revised MS the authors reanalyzed the COI dataset using a new software. The description and interpretation of the COI tree is still inadequate providing the impression that the authors are not familiar with molecular phylogenetic methods. The authors indicated values of three statistical tests corrersponding to nodes on thier tree. The authors wrote that "These two subclades were supported by 87.6/0.943/62", however this is not what can be seen in the tree. In the tree, the subgenus Bryobia is very poorly supported (marginally or even non-supported) (63/0.6/51) and the monophyly of the second subgenus was not recovered at all. The authors are requested to describe their analysis correcrtly and avoid misinterpretation of the results. Therefore the section 3.4.2. needs major revision. Please, do not speculate on Wolbachia in this section, Wolbachia does not influence your results at all, this effect can be observed only in large phylogenies and Wolbachia would not "spoil" true large clades corresponding to large suprageneric host lineages.

line 354 TNT-tree analysis by new technology - please, explain what do you mean saying "new technology"

QUESTION. The authors wrote in their reply to reviewer comment#4: "The Figure 4-old included morphological analysis of most species of the Bryobia genus and three species of the closely related genus Pseudobryobia as outgroup. The Figure 4-new included only the type species of the proposed subgeneric divisions and that of an outgroup. " Actually, this information is absent in the MS, so it is not clear that only 4 species were included in the analysis.  Please, explain your morphological analysis correctly. Finally, performing analysis using only 4 species is incorrect, because after the analysis you extrapolate the results on many other species which were not included in ther analysis (like it was done in the old version). Please, revise the part of the MS concerning the morphological nalysis. Additionally, please explain why the results of Fig4old vs Fig4new are so different?

Comments on the Quality of English Language

moderate

Author Response

Review Round III

Author’s response to Reviewer Comments

Comment#1 In the revised MS the authors reanalyzed the COI dataset using a new software. The description and interpretation of the COI tree is still inadequate providing the impression that the authors are not familiar with molecular phylogenetic methods. The authors indicated values of three statistical tests corrersponding to nodes on thier tree. The authors wrote that "These two subclades were supported by 87.6/0.943/62", however this is not what can be seen in the tree. In the tree, the subgenus Bryobia is very poorly supported (marginally or even non-supported) (63/0.6/51) and the monophyly of the second subgenus was not recovered at all. The authors are requested to describe their analysis correcrtly and avoid misinterpretation of the results. Therefore the section 3.4.2. needs major revision. Please, do not speculate on Wolbachia in this section, Wolbachia does not influence your results at all, this effect can be observed only in large phylogenies and Wolbachia would not "spoil" true large clades corresponding to large suprageneric host lineages.

Response: The authors would like to thank the reviewer for pointing this out. The results have been modified in the section 3.4.2. The speculations on Wolbachia were removed during the first round of review and are no longer present in the revised manuscript.

Comment#2 line 354 TNT-tree analysis by new technology - please, explain what do you mean saying "new technology

Response: If your question is understood correctly, it is the complete name of the software. On the other hand, the explanation for new technology goes with the use of algorithms for analysis. It has been provided in the methodology section of the revised version.

Comment#3 QUESTION. The authors wrote in their reply to reviewer comment#4: "The Figure 4-old included morphological analysis of most species of the Bryobia genus and three species of the closely related genus Pseudobryobia as outgroup. The Figure 4-new included only the type species of the proposed subgeneric divisions and that of an outgroup. " Actually, this information is absent in the MS, so it is not clear that only 4 species were included in the analysis.  Please, explain your morphological analysis correctly. Finally, performing analysis using only 4 species is incorrect, because after the analysis you extrapolate the results on many other species which were not included in ther analysis (like it was done in the old version). Please, revise the part of the MS concerning the morphological nalysis. Additionally, please explain why the results of Fig4old vs Fig4new are so different?

Response: The first Figure 4 (fig4 old) present in the manuscript at the time of submission, was replaced by fig4new during the first round of review. The reason has been answered previously. Considering the point, as raised by reviewer here, the analysis was re-done (including 84 species with distinct/nonviable character states) during the 2nd round of review and fig4new was again replaced by new tree. The New tree was the result of new technology search using modern techniques while fig4new and fig4old were contructed by multivariate analysis, considered as old method. Hence, there are obvious differences.

It is to bring to attention that both the old and new figure 4 were removed in the previous and were replaced with updated phylogenetic after re-analysis in the revised version.

Round 4

Reviewer 2 Report

Comments and Suggestions for Authors

The authors carefully revised the MS. There are several typos and extra dots. The authors are requested carefully read through the entire MS and revise these small flaws.

Please, revise the sentences containing mentioning the "New Techology". TNT stands for "Tree analysis using New Technology". It is based on parsimony analysis, no other special "new technologies". Please check here:

https://www.lillo.org.ar/phylogeny/tnt/

TNT stands for "Tree analysis using New Technology". It is a program for phylogenetic analysis under parsimony

Comments on the Quality of English Language

minor

Author Response

Review Round IV

Author’s Response to Reviewer Comments

Comment#1 The authors carefully revised the MS. There are several typos and extra dots. The authors are requested carefully read through the entire MS and revise these small flaws.

Response: The authors have re-read the whole manuscript. The typos and extras have been removed, and English Language and Grammar proofing has been completed through Grammarly (Licenced Software).

Comment#2 Please, revise the sentences containing mentioning the "New Techology". TNT stands for "Tree analysis using New Technology". It is based on parsimony analysis, no other special "new technologies". Please check here:https://www.lillo.org.ar/phylogeny/tnt/

TNT stands for "Tree analysis using New Technology". It is a program for phylogenetic analysis under parsimony

Response: The sentence has been rephrased in the revised version.
